# OpenReview forum: "Learn Your Reference Model for Real Good Alignment"
_ICLR.cc/2025/Conference — ICLR 2025 Poster_

### Official Review · Reviewer_JVHU · 2024-10-25

**Soundness:** 3
**Presentation:** 2
**Contribution:** 3
**Rating:** 6
**Confidence:** 4

**Summary:**

The paper, titled Learn Your Reference Model for Real Good Alignment, presents a new method for aligning large language models (LLMs) by introducing the Trust Region (TR) approach. This method dynamically updates the reference policy throughout training to reduce overoptimization issues in offline alignment methods. Three key variations are proposed: TR-DPO, TR-IPO, and TR-KTO, which improve upon existing alignment methods by better managing the divergence from the reference policy.

**Strengths:**

+ The introduction of Trust Region methods to continuously update the reference policy reduces overoptimization issues, maintaining model performance even when deviating from the initial reference model.
+ The methods outperform existing baselines in tasks like summarization and dialogue alignment, showing significant improvements in win rates on benchmarks such as AlpacaEval 2 and Arena-Hard.

**Weaknesses:**

- The paper also lacks defense performance against well-known jailbreak methods, such as GCG and GPTFuzzer.
- The paper does not deeply analyze the potential trade-offs between this additional cost and the performance gains.
- More on questions.

**Questions:**

The paper relies heavily on automatic evaluation methods using GPT-4 as a proxy for human judgment, which could raise concerns about the robustness of the evaluation.

---

> ### Author Response · Authors · 2024-11-15
>
> First of all, we would like to sincerely thank you for taking the time and effort to review our paper. Your thoughtful and constructive feedback provides us with valuable opportunities to clarify and improve our work. We deeply appreciate your recognition of our contributions, such as the introduction of Trust Region methods to mitigate overoptimization, the significant performance improvements demonstrated on benchmarks like AlpacaEval 2.0, and the novelty of our approach in addressing alignment issues in summarization and dialogue tasks.
>
> We have carefully addressed the weaknesses and questions raised, as detailed below.
>
> **Comments on Weaknesses**
>
> - **Jailbreak robustness**
>
>     We agree that robustness against jailbreak methods is an important aspect of evaluating the reliability of LLMs. In this work, we focused on the alignment of models with human preferences, following the research direction and experimental setups of prior studies [e.g., 1, …, 13] that also aim to address alignment challenges. These setups are specifically designed to analyze the effectiveness of alignment strategies rather than robustness against jailbreak attacks. While this is an important topic, it lies outside the primary scope of our study, as our goal was to contribute to advancing alignment methods. We appreciate the reviewer bringing this up and consider it a valuable direction for future research.
>
> - **Cost-performance trade-off**
>
>     We appreciate the reviewer bringing up the consideration of trade-offs between computational cost and performance gains. However, the comment lacks specific examples or metrics to clarify what aspects of this analysis they find insufficient. To address this, we provide a detailed summary of the computational costs and performance improvements evaluated in our work.
>
>     The additional computational cost of our proposed method compared to traditional approaches is at most $O(n)$, where $n$ is the number of model parameters, when using soft updates in each training step. For the evaluation, we measured the training times of various configurations of our method and baseline approaches on the UltraFeedback dataset, as described in our paper (line 286). Each model was trained for a single epoch to ensure consistency. The configurations included Llama 3 8B Base with soft updates ($\alpha = 0.8$) and hard updates ($\tau = 32$), as well as baseline methods such as DPO, IPO, and KTO. For TR-KTO, we specifically used $\alpha = 0.6$ for soft updates and $\tau = 64$ for hard updates, as outlined best configuration in Table 6 of our work.
>
>     On average, we observed a moderate increase in training time for our TR-based methods compared to their respective baselines:
>
>     - For DPO, the training time increased by **3.99%** (soft updates) and **2.87%** (hard updates).
>     - For IPO, the increase was **6.93%** (soft updates) and **3.32%** (hard updates).
>     - For KTO, the increase was **7.44%** (soft updates with $\alpha = 0.6$) and **3.65%** (hard updates with $\tau = 64$).
>
>     Despite the additional computational cost, the improvements in AlpacaEval 2.0 benchmark performance are substantial, as reported in the current version of our work (line 963). For example:
>
>     - DPO achieved an average improvement of **71.82%** (soft updates) and **74.72%** (hard updates) on .
>     - IPO showed an increase of **85.76%** (soft updates) and **76.94%** (hard updates).
>     - KTO improved by **30.93%** (soft updates with $\alpha = 0.6$) and **28.65%** (hard updates with $\tau = 64$).
>
>     These results underline the efficiency of our method, demonstrating that the performance gains significantly outweigh the modest increase in computational requirements.
>
>     We will include this detailed analysis in the rebuttal revision version of the paper to provide further transparency and clarity on the cost-performance trade-offs associated with our approach.

---

> ### Author Response · Authors · 2024-11-15
>
> **Comments on Questions**
>
> - **Use of GPT-4 and automatic evaluation**
>
>     It is true that we employ GPT-4 and AlpacaEval 2.0 as part of our evaluation framework. AlpacaEval 2.0, which demonstrates the highest correlation with the LLM Arena benchmark (https://huggingface.co/spaces/lmarena-ai/chatbot-arena-leaderboard), is specifically used to evaluate model performance. This is explicitly described in the Discussion section, where we also acknowledge the potential bias this might introduce. However, this approach has become the de facto standard in the field, as evidenced by its widespread use in numerous works published at A/A* conferences [e.g., 1, 7, 12, 14, 15, 16, 17, 18]. By adopting this evaluation strategy, we ensure comparability and consistency with state-of-the-art methods in the field.
>
>
> **Conclusion**
>
> We deeply appreciate the reviewer’s constructive feedback, which has helped us to clarify and strengthen our work. We hope that the additional details provided address the reviewer’s concerns and highlight the contributions of our work more clearly. Given the novelty of our method, its demonstrated performance improvements, and its alignment with current evaluation standards, we kindly request the reviewer to reconsider their rating.
>
> **References:**
>
> [1] Rafailov R. et al. Direct preference optimization: Your language model is secretly a reward model
>
> [2] Zhao Y. et al. Slic-hf: Sequence likelihood calibration with human feedback
>
> [3] Liu T. et al. Statistical rejection sampling improves preference optimization
>
> [4] Munos R. et al. Nash learning from human feedback
>
> [5] Azar M. G. et al. A general theoretical paradigm to understand learning from human preferences
>
> [6] Ethayarajh K. et al. Kto: Model alignment as prospect theoretic optimization
>
> [7] Meng Y., Xia M., Chen D. Simpo: Simple preference optimization with a reference-free reward
>
> [8] Melnyk I. et al. Distributional Preference Alignment of LLMs via Optimal Transport
>
> [9] D'Oosterlinck K. et al. Anchored Preference Optimization and Contrastive Revisions: Addressing Underspecification in Alignment
>
> [10] Wang R. et al. ASFT: Aligned Supervised Fine-Tuning through Absolute Likelihood
>
> [11] Chen H. et al. Noise contrastive alignment of language models with explicit rewards
>
> [12] Hong J., Lee N., Thorne J. Reference-free monolithic preference optimization with odds ratio
>
> [13] Xu H. et al. Contrastive preference optimization: Pushing the boundaries of llm performance in machine translation
>
> [14] Ding N. et al. Enhancing chat language models by scaling high-quality instructional conversations
>
> [15] Park R. et al. Disentangling length from quality in direct preference optimization
>
> [16] Rafailov R. et al. Scaling laws for reward model overoptimization in direct alignment algorithms
>
> [17] Yuan W. et al. Self-rewarding language models
>
> [18] Xiong W. et al. Iterative preference learning from human feedback: Bridging theory and practice for rlhf under kl-constraint

---

> ### Comment · Reviewer_JVHU · 2024-11-21
>
> Thanks for your response, and I think the advanced jailbreak method is needed to prove the method's robustness. If you provide these parts for experiments, I am willing to improve my score. Good luck with your rebuttal.

---

> ### Author Response · Authors · 2024-11-24
>
> We are pleased that we were able to clarify some aspects of our work that were unclear to you. We are also grateful for the opportunity to improve the evaluation. We conducted the experiments you requested, specifically, we measured our models on two jailbreak benchmarks: GPTFuzzer [1] and ReNeLLM [2]. The selection of these two methods is based on their reasonably high Attack Success Rate (ASR) (approximately the same for both methods) with a relatively short operation time (Figure Appendix 4 from paper [3]). We used the EasyJailbreak [3] library to test our models. We used the setup from the original papers describing the Jailbreak method.
>
> **GPTFuzz**
>
> - We used a list of questions expected to elicit bad response from the GPTFuzz's original library (https://github.com/sherdencooper/GPTFuzz/blob/master/datasets/questions/question_list.csv) as instructions.
> - gpt-4-1106-preview was used for mutation of templates.
> - Roberta, trained by GPTFuzz authors (https://huggingface.co/hubert233/GPTFuzz), was used as the model to evaluate the responses of the targeted model. This model showed the best accuracy among other models (even better than GPT-4 according to their data).
> - We set a limit on the number of requests to the targeted model at 10,000, the number of jailbreak templates at 1,000, the number of unjailbroken requests at 10,000, and also the iteration limit at 100. If one of these limits was reached, the search algorithm would terminate.
> - We measured two metrics in the same way as in the paper. In ASR top 1, the best template by the number of broken questions is chosen and the percentage of broken questions is measured. In ASR top 5, five best templates by the number of broken questions are chosen and the percentage of questions broken by at least one of the templates is measured.
>
> **ReNeLLM**
>
> - For the instructions that provoke the model into giving a bad response, we used instructions from AdvBench [4].
> - gpt-4-1106-preview was used both as the attacking model and to evaluate the responses of the targeted model.
>
> **Our Models**
>
> We selected the best hyperparameters $\alpha$ and $\tau$ for our methods according to Appendix Table 7 in our paper. In addition, for each method we took the optimal value for the $\beta$ parameter according to our new Pareto Front analysis for Llama-3-Base models. The Pareto Front analysis is presented in the figure: https://pdfupload.io/docs/4007e828 (anonymous link).
>
> The Jailbreak benchmark results are presented in the table.
>
> | METHOD  | $\beta$  | GPTFuzz Top-1 ASR (%) | GPTFuzz Top-5 ASR (%) | ReNeLLM GPT ASR (%) |
> | --- | --- | --- | --- | --- |
> | DPO | 0.005 | **97** | 100 | 100 |
> | TR-DPO ($\alpha$ = 0.8) | 0.03 | 99 | **98** | 100 |
> | TR-DPO ($\tau = 32$) | 0.03 | 98 | **98** | 100 |
> | IPO | 0.001 | 99 | **99** | 100 |
> | TR-IPO ($\alpha = 0.8$) | 0.01 | 99 | 100 | 100 |
> | TR-IPO ($\tau = 32$) | 0.01 | **97** | **99** | 100 |
> | KTO | 0.03 | 99 | 100 | 100 |
> | TR-KTO ($\alpha = 0.6$) | 0.05 | 96 | 100 | 100 |
> | TR-KTO ($\tau = 256$) | 0.05 | **94** | **99** | 100 |
>
> As can be seen from the table, according to the GPTFuzz benchmark, DPO outperforms our modification when using top 1 template, however, our modifications are more resistant when using top 5 templates. TR-IPO$^\tau$ performs better than the other two when calculating top 1 ASR and performs on par with standard IPO when calculating top 5 ASR. TR-KTO$^\tau$ performs better than the other two methods according to both metrics. ReNeLLM was able to hack all models in 100% of cases. We attribute this to the use of GPT-4 for evaluating model responses instead of Roberta in this case.
>
> With all due respect, we still believe that resistance to attacks is an orthogonal issue to the overoptimization problem we study in our work, as can be seen from the results. Resilience to attacks should be accomplished using specific methods aimed at addressing this particular issue. However, we hope that we have satisfied your curiosity about how resilient our method is to jailbreak attacks, and we hope that you will raise your score.
>
> **References:**
>
> [1] Yu et al, GPTFUZZER: Red Teaming Large Language Models with Auto-Generated Jailbreak Prompts
>
> [2] Ding et al, A Wolf in Sheep's Clothing: Generalized Nested Jailbreak Prompts can Fool Large Language Models Easily
>
> [3] Zhou et al, EasyJailbreak: A Unified Framework for Jailbreaking Large Language Models
>
> [4] Zou et al, Universal and Transferable Adversarial Attacks on Aligned Language Models

---

> > ### Comment · Reviewer_JVHU · 2024-11-25
> >
> > Thanks for your update experiment, I update my review.

---

> > > ### Author Response · Authors · 2024-11-25
> > >
> > > We sincerely appreciate that you have raised your rating in light of our updates.
> > >
> > > We have endeavored to ensure that our work is systematically sound and makes a substantial contribution to our scientific community. As such, we have conducted all the experiments you suggested, and we have addressed your concerns in a thorough and detailed manner, namely, we analyzed defense performance against well-known jailbreak methods, and also provided performance measurement figures for our method. Your evaluation, however, indicates that our work still marginally falls below the acceptance threshold, which we find slightly bewildering.
> > >
> > > In this context, we kindly ask you to provide more specific feedback on other elements of our paper that might still be lacking or need improvements. Are there any additional experiments or areas we should delve into, that we might have overlooked?
> > >
> > > If upon reflection you believe that we have addressed all your concerns and there are no more areas left to improve, we would be grateful if you would reconsider the assessment of our paper in our favor.

---

> > > > ### Author Response · Authors · 2024-11-27
> > > >
> > > > Thank you for raising your score after our updates and additional experiments. However, we’re a bit surprised that the overall rating remains a 5 (the same as before your downgrading to 3), even after we addressed the specific concerns raised in your review. We’ve added Appendix G with detailed jailbreak results and emphasized their importance, as well as provided additional details on the computational efficiency of our method in Appendix D.3, fully addressing your feedback.
> > > >
> > > > If there are any remaining issues or areas you feel are still lacking, please let us know so we can address them. Otherwise, we kindly ask you to consider raising your score.

---

### Official Review · Reviewer_V83C · 2024-10-29

**Soundness:** 3
**Presentation:** 3
**Contribution:** 2
**Rating:** 5
**Confidence:** 4

**Summary:**

The paper introduces a novel paradigm for aligning Large Language Models through offline methods. The authors propose Trust Region (TR) methods, including TR-DPO, TR-IPO, and TR-KTO, which dynamically update the reference policy during the training process. The paper claims that these methods effectively mitigate overoptimization, allowing models to maintain strong performance even with significant deviations from the initial reference policy. The efficacy of these approaches is demonstrated through toy examples and specific tasks such as dialogue and summarization, where they outperform conventional methods. Additionally, the paper reports significant improvements on general-purpose assistant benchmarks using the Llama3 model.

**Strengths:**

1. The paper introduces a novel Trust Region (TR) approach that significantly mitigates the issue of reward overoptimization in offline learning of Large Language Models. By dynamically updating the reference policy during the training process, the proposed method potentially offers a more robust alignment technique.

2. The paper conducts a comprehensive set of experiments across various tasks and models. The authors have meticulously detailed their experimental procedures and published their parameters, which not only supports the credibility of their findings but also allows for reproducibility by other researchers in the field.

3. The paper demonstrates a strong understanding of the underlying issues in offline alignment methods and provides a clear motivation for the proposed Trust Region methods. The authors have effectively communicated the problem of overoptimization and how their approach can mitigate this through updating the reference policy, which is a conceptual strength of the paper.

**Weaknesses:**

1. The methods proposed in this paper—soft and hard updates for the reference policy—are fundamentally extensions of existing approaches rather than novel contributions. Soft updates are commonly used in reinforcement learning and do not introduce new mechanisms or insights; similarly, hard updates reflect established practices without significant advancement. Moreover, while the introduction of Trust Region (TR) methods suggests a framework, it does not provide a substantial departure from conventional strategies, as the core mechanics remain similar.

2. The claim that this update will "increase the curvature of the loss landscape" is not backed by theoretical support. The mechanism for alleviating overoptimization remains unclear, and the suggestion that moving away from the SFT policy is "not inherently bad" could be misleading, as potential negative consequences are not addressed.

3. The paper relies heavily on automatic evaluation methods, such as GPT-4, to assess model performance. Using a single model (like GPT-4) as a judge in evaluations could introduce bias, as it may not fully capture the nuances of human evaluation. The paper would be strengthened by including human evaluation or a discussion on the reliability of the evaluation methods used.

**Questions:**

1. What is the mechanism for alleviating overoptimization? Please investigate it more clearly.

---

> ### Author Response · Authors · 2024-11-21
>
> Thank you very much for your thorough review and for highlighting the strong points of our work, namely the extensive experimental setup, reproducibility of results, and the clarity of motivation for our method and its essence. We appreciate the opportunity to address your concerns and to comment on the weaknesses you've highlighted in our paper.
>
> **Weaknesses:**
>
> 1. Our aim was to investigate the application of update techniques to the reference policy in Direct Alignment Algorithms (DAA). To test our primary hypothesis that updating the reference policy enhances the quality of alignment, we selected understandable and simple update methods from RL to avoid making testable hypothesis multifaceted. The term 'new paradigm' in the abstract refers to the fact that until now, no one has updated the reference policy in DAA. Our research indicates that a dynamic reference policy performs better than a static one, contributing insights about the workings of base methods (DPO, IPO, KTO) and opening a perspective for further research on this issue. We understand that the usage of the phrase 'new paradigm' could lead to confusion, and we will remove it accordingly.
> 2. We do not provide a rigorous proof to the fact that updating the reference policy can decelerate the overoptimization process. Rather, we only offer intuition on how it impacts the training process, namely, it periodically increases the scale of the gradient, which increases the probability of the model escaping from the overoptimization zone (Section 3.1). Additionally, we provide empirical validation of slowing down overoptimization during the training process through extensive experiments, starting with a toy MDP example and ending with benchmark measurements for the general assistant task.
> 3. We want to draw attention to the fact that this is explicitly addressed in the Discussion section of our paper, where we highlight the potential bias that might be introduced by the techniques we used. Still, the chosen methods provide a high degree of reliability as evidenced by their high correlation with human evaluation. For instance, AlpacaEval 2.0, which has shown the highest correlation with the LLM Arena (https://huggingface.co/spaces/lmarena-ai/chatbot-arena-leaderboard) — a human annotated benchmark, is used to assess model performance. While we appreciate your recommendation to include human evaluation or a discourse on the reliability of the evaluation methods used, we have chosen to use these evaluation strategies because they have become the de facto standard in the field [1, 2, 3, 4, 5, 6, 7]. By utilizing these methods, we aim to ensure that our work can be directly compared to, and is consistent with, state-of-the-art techniques in our field.

---

> ### Author Response · Authors · 2024-11-21
>
> **Question:**
>
> We consider the Hessian of the DPO loss function, which looks like this:
>
> $
> \nabla^2_\theta \mathcal{L}_ {\text{DPO}}(\pi_\theta, \pi_{\text{ref}}) =
> \mathbb{E}_ {(x, y_w, y_l) \sim \mathcal{D}} \left[\sigma(s) \nabla^2_\theta s + \sigma(s) \big(1 - \sigma(s)\big) \nabla_\theta s \big(\nabla_\theta s\big)^\top\right],
> $
>
> where $s = \beta\log\frac{\pi_\theta(y_l|x)}{\pi_\theta(y_w|x)} - \beta\log\frac{\pi_{\text{ref}}(y_l|x)}{\pi_{\text{ref}}(y_w|x)}$. During the model training, the value of $s$ generally decreases and moves towards values smaller than zero for the methods with static reference model. This is because $\beta\log\frac{\pi_{\text{ref}}(y_l|x)}{\pi_{\text{ref}}(y_w|x)}$ is fixed, and the value of $\beta\log\pi_\theta(y_l|x)$  is less than the value of $\beta\log\pi_\theta(y_w|x)$  and decreases at a faster rate. This means that the value of $\sigma(s)$ tends to zero. Since this value is involved in both terms (the second term also has a factor $(1 - \sigma(s))$ which is limited to 1), the value of the entire Hessian tends to 0. This means that if the model has entered the over-optimization region over the training time, it will be difficult for it to get out of this process, since the gradient hardly changes (the Hessian is close to zero).  Our method allows us to update the value of $s$, for example, with a hard update it becomes equal to 0 every time in some number of steps, since $\pi_\mathrm{ref}$ becomes equal to $\pi_\theta$. $\sigma(s)$ becomes equal to 0.5, making the gradient larger. This makes it easier for the optimization process to push the probabilities chosen up and rejected down, and the probability of getting out of overoptimization is higher.
>
> We hope that our explanations have clarified some unclear aspects of our proposed method. Your questions and feedback are immensely valuable to us, and we genuinely appreciate the opportunity to clarify our work based on your insights. If you feel that we have addressed all of your concerns, and you are satisfied with the discussions so far, we would very much appreciate it if you could consider improving the rating of our work.
>
> **References**
>
> [1] Meng Y., Xia M., Chen D. Simpo: Simple preference optimization with a reference-free reward
>
> [2] Hong J., Lee N., Thorne J. Reference-free monolithic preference optimization with odds ratio
>
> [3] Ding N. et al. Enhancing chat language models by scaling high-quality instructional conversations
>
> [4] Park R. et al. Disentangling length from quality in direct preference optimization
>
> [5] Rafailov R. et al. Scaling laws for reward model overoptimization in direct alignment algorithms
>
> [6] Yuan W. et al. Self-rewarding language models
>
> [7] Xiong W. et al. Iterative preference learning from human feedback: Bridging theory and practice for rlhf under kl-constraint

---

> > ### Comment · Reviewer_V83C · 2024-11-22
> > **Thank you for your clarification**
> >
> > Thanks for your clarification, which has partly addressed my concerns. On the other hand, the novelty issue remains. I tend to keep my score.

---

> > > ### Author Response · Authors · 2024-11-22
> > >
> > > We are pleased that we were able to address your concerns. Our aim was to investigate the effect of dynamic updates of the reference policy in DAA — an area not previously explored — rather than to propose a new update mechanism. If you believe that the novelty of this work is in question, could you please provide references to works that have done the same?

---

> > > > ### Comment · Reviewer_V83C · 2024-11-26
> > > > **I still believe that the novelty of this approach is limited**
> > > >
> > > > Thank you for your valuable comments. I still believe that the novelty of this approach is limited. In my opinion, this paper identifies an intermediate state between "fixed reference model" and "periodically updated reference model." This method is too common in traditional RL algorithms. Of course, I acknowledge other aspects of this paper, including the extensive experiments and in-depth analysis of the method's effectiveness. I just think it falls below the acceptance threshold for ICLR.

---

> ### Author Response · Authors · 2024-11-26
>
> Thank you for your continued engagement and thoughtful feedback. We appreciate your recognition of our experimental rigor and would like to directly address your concerns about the novelty of our approach.
>
> **Introducing Trust Region Methods to Offline DAA:**
>
> While Trust Region (TR) methods are well-known in online reinforcement learning, their relevance to offline DAA was previously unexplored. Our work is the first to adapt TR methods to address the specific challenge of overoptimization in offline DAA.
>
> In online RL, TR methods are used primarily to stabilize policy updates. However, they had not been connected to the problem of overoptimization in offline settings. Our work establishes this connection and demonstrates that periodic updates to the reference policy using TR methods effectively resolve overoptimization without altering the loss function. This is a departure from prior approaches, which relied on loss modifications to tackle this problem. By introducing this mechanism, we provide the field with a novel tool for alignment and set the stage for broader adoption of TR methods in offline settings.
>
> **Understanding Overoptimization in DAA:**
>
> In addition to adapting TR methods, our work provides new theoretical insights into the phenomenon of overoptimization in DAA. Specifically, we analyze the Hessian of the DPO loss function to demonstrate how a static reference policy leads to optimization stagnation by reducing gradient magnitudes over time. This stagnation traps the model in overoptimized regions, hindering its ability to escape.
>
> Dynamic updates to the reference policy, as proposed in our method, counteract this stagnation by periodically resetting optimization dynamics. This ensures larger gradients and enables more effective training. Importantly, our analysis goes beyond practical improvements—it offers the theoretical explanation of how overoptimization arises in DAA and how it can be mitigated through changes in the optimization process rather than the loss function itself.
>
> We hope this clarification highlights the novelty and significance of our contributions. Our goal is to advance the field by introducing and thoroughly investigating an approach that, while inspired by concepts in RL, is novel in its application to offline LLM alignment through DAA. This goes beyond improving scores—our findings enhance the community's understanding of how DAA methods function and offer guidance on how they can be further improved. The existence of Trust Region (TR) methods alone did not imply their applicability to DAA. Merely showing that they improve performance would not justify their adoption. Instead, our work highlights why TR methods are effective in this context, providing the theoretical and empirical foundation for their use in DAA.
>
> We are open to any further questions or discussions you may have.

---

### Official Review · Reviewer_xzHv · 2024-10-31

**Soundness:** 2
**Presentation:** 4
**Contribution:** 3
**Rating:** 8
**Confidence:** 5

**Summary:**

The authors proposed a new offline paradigm for aligning large language models using preference optimization methods. They explored the overoptimization problem in state-of-the-art preference optimization methods and showed that the current methods suffer from this issue. To overcome this problem, they updated the reference models with the weight of the policy model on safe and hard settings. They showed that this method outperforms the vanilla across different metrics.

**Strengths:**

The main strengths of this paper are comprehensive experiments and analysis of various benchmarks and models. Also, they achieved impressive performance with a simple change on the optimization part, which means updating the weight of the reference model during the optimization part.

**Weaknesses:**

This paper showed impressive improvement on different benchmarks; however, I have multiple concerns about this paper.


1. **Lack of generalizability**. The authors called their method a new paradigm for offline alignment techniques. However,  some new methods, like SimPO, CPO, and TPO, achieve impressive performance compared with DPO, IPO, and KTO by removing the reference model. So, I am curious how we can call it a new paradigm.

2. **Lack of novelty**. The ORPO method, motivated by maximizing the likelihood of a bad response during the SFT, proposed another method to overcome the issue. They showed that this method can also resolve the overoptimization, and the log probability of the chosen response increases during the optimization (refer to Figure 7 in the ORPO paper).


3. **Lack of explanation**. In paper, the authors to satisfy their hypothesis, they did some experiments on OOD trajectory probability, as shown in Figure 2. However, this experiment is not clear and I am wondering which data is used as out-of-distribution (OOD).



4. **Huge discrepancies and a lack of exploration of hyper-parameters for different methods**. The results reported for LLama3-Base on AlpacaEval 2 and ArenaHard for DPO, IPO, and KTO in the SimPO paper and this paper are very different. The discrepancy on ArenaHard for IPO is more than 11%, and for DPO, it is more than 5%. I refer the authors to Table 4 in the SimPO paper []. Also, It seems the authors didn’t choose the best hyperparameter for DPO. In Appendix D.1 they selected 1.0e-6 for llama as the learning rate for DPO, while learning 5.0e-7 is the best for DPO. We refer to the SimPO paper because the DPO model they fine-tuned has better performance than this paper.


5. **Lack of efficiency analysis**. This method needs to keep the reference model in Memory during the optimization. To improve the efficiency of the DPO, KTO, and IPO, the value of the reference model can be calculated before training. So, I think efficiency analysis shows different aspects of this method. Reporting the Peck GPU memory and run-time is helpful.


6. Recent papers like Zephyr and SimPO showed that preference optimization methods not only need more steps for optimization but also, after one epoch, will be overfitting on data. However, the proposed method requires large steps for hard update settings.

---
SimPO: https://arxiv.org/abs/2405.14734

CPO: https://arxiv.org/abs/2401.08417

TPO: https://arxiv.org/abs/2405.16681

ORPO: https://arxiv.org/abs/2403.07691

Zephyr: https://arxiv.org/abs/2310.16944

**Questions:**

I have a couple of questions and suggestions. I appreciate the authors answering the following:


1. The same dataset generated by llama3-instruct-8b was prepared by the SimPO paper before. I suggest the authors to fine-tuned the LLaMA3-8b-SFT (https://huggingface.co/princeton-nlp/Llama-3-Base-8B-SFT) with their method on this UltraFeedback (https://huggingface.co/datasets/princeton-nlp/llama3-ultrafeedback) and report the AlpacaEval and ArenaHard.

2. most of the analysis on KL divergence is on Pythia 2.8. I am happy to see some analysis on LLama3 models to verify the observation.

3. I suggest the authors evaluate the llama3 models on the MixEval (https://github.com/Psycoy/MixEval/?tab=readme-ov-file) benchmark, too.

---

> ### Author Response · Authors · 2024-11-24
>
> First of all, we would like to sincerely thank you for taking the time and effort to review our paper. Your feedback is greatly appreciated and helps us refine our work. We are especially grateful for your recognition of the strengths of our paper, including the comprehensive experiments and analysis across various benchmarks and models, as well as the impressive performance achieved through a simple yet effective adjustment in the optimization process by updating the reference model's weight.
>
> We have carefully addressed the weaknesses and questions raised, as detailed below.
>
> **Comments on Weaknesses**
>
> - **Lack of generalizability**
>
>     We agree that there are alternative approaches, such as SimPO, CPO, and TPO, which achieve impressive results by removing the reference model. However, in this paper we consider the class of methods where the reference policy is present. Results below, demonstrate that dynamically updating the reference policy is a viable and effective approach in offline alignment achieving comparable results with SimPO method. Previously, the techniques of updating the reference policy have been used primarily in online alignment setups (e.g., Nash Learning approaches like Nash-MD [1] and DNO [2]). Our proposed approach extends this idea to offline settings and is supported by the minimal changes required to existing offline frameworks, while yielding significant improvements in performance.
>
>     Additionally, we acknowledge that the use of the term "new paradigm" in line 14 of our paper may be misleading. We will revise this wording to better reflect our intention, which is to propose an enhancement to existing reference policy based methods.
>
> - **Lack of novelty**
>
>     We greatly appreciate your observations regarding ORPO and its approach to addressing overoptimization. However, our work differs significantly from ORPO in both motivation and implementation. ORPO addresses overoptimization by maximizing the likelihood of desirable responses during SFT. A similar effect can be achieved by simply adding a cross-entropy loss to the DPO/IPO/KTO losses, as done in LLama 3.1 [3], CPO [4] and Iterative RPO [5]. In contrast, our work explores a different approach to overcoming overoptimization. While adding a cross-entropy loss to the DAA framework is a straightforward and simple method to mitigate overoptimization, we demonstrate a fundamentally different way of addressing this problem. By working within the DAA framework, we show that the reference policy can be dynamically updated and effectively utilized, providing impactful insights into the area of overoptimization.
>
>     To further clarify, we demonstrate the motivation for our method by analyzing the Hessian of the DPO loss function, providing insights into the optimization dynamics. This is complemented by a toy example [6], which allows for a detailed examination of the method's nuances in a controlled setting. Additionally, we conduct Pareto front evaluations that highlight superior performance across KL divergence and generation quality metrics compared to baseline approaches.
>
>     This analysis underscores the distinctiveness and practical value of our method. Moreover, as mentioned earlier, our work demonstrates that the Trust Region (TR) approach, previously limited to online alignment settings, can be effectively extended to offline alignment. By showcasing this adaptability, we believe our work provides a meaningful contribution to the field.
>
> - **Lack of explanation**
>
>     The toy example setup is outlined in Appendix B. We duplicated the setup completely from Rafailov et al [6]'s work. In the MDP, there are only 9 trajectories (sequences of length 3, with 3 possible tokens at each position). 3 of these trajectories are used as the SFT dataset. An RNN is trained on this dataset for 50 epochs. 2 of these trajectories are used as a pair of preferences (the dataset consists of a single pair), on which the RNN is trained for 200 epochs. These 2 trajectories, used in training the model with the preference loss function, are considered ID, while the remaining 7 (including the one that was in the SFT, but did not involved into the pair) are considered OOD.

---

> ### Author Response · Authors · 2024-11-24
>
> - **Huge discrepancies and a lack of exploration of hyper-parameters for different methods**
>
>     We would like to address the discrepancies and clarify the reasons behind the differences in hyperparameter choices between our work and the SimPO paper.
>
>     In SimPO, the DPO method used a $\beta$ value of 0.01, whereas in our work, we selected $\beta$ = 0.05 based on Pareto front evaluations (Figure 5 of our paper). These evaluations, performed on the Pythia 2.8 model, demonstrated that $\beta$ = 0.05 yields better performance. This choice was consistent across our experiments, ensuring alignment with our stated evaluation criteria. Moreover, for IPO and KTO, in both the Llama3-Base and Llama3-Instruct setups on AlpacaEval 2.0 LC, our results show higher performance compared to those reported in the SimPO paper.
>
>     To further address your suggestion, we extended our experiments to include results for DPO with different learning rates, specifically comparing the values of 5e-7 (as suggested in SimPO) and 1e-6 for Llama3-Base. For Llama3-Instruct, we will also include results with 7e-7, which the SimPO authors claim as a better-performing value (https://github.com/princeton-nlp/SimPO). Our preliminary results indicate that for both setups, the learning rate of 1e-6 yields better performance in terms of both LC and WR.
>
>     **Results for Llama3-Base**
>
>     | Method | Lr | Beta | LC (std) | WR (std) | Length |
>     | --- | --- | --- | --- | --- | --- |
>     | DPO | 5e-07 | 0.01 | 21.07 (0.83) | 18.16 (1.17) | 1697 |
>     | DPO | 1e-06 | 0.01 | 23.58 (0.84) | 20.87 (1.20) | 1723 |
>
>     **Results for Llama3-Instruct**
>
>     | Method | Lr | Beta | LC (std) | WR (std) | Length |
>     | --- | --- | --- | --- | --- | --- |
>     | DPO | 7e-07 | 0.01 | 39.54 (0.82) | 42.2 (1.44) | 2128 |
>     | DPO | 1e-06 | 0.01 | 41.15 (0.81) | 45.68 (1.46) | 2216 |
>
>     To ensure our results comprehensively address this issue, we will also include additional analyses of the optimal $\beta$ values for each method (DPO, IPO, KTO, TR-methods) as shown in the Pareto fronts discussed in response to Questions 2. Notably, we observe that the SimPO paper also did not explore optimal $\beta$ values for some baseline methods, which underscores the value of our work in systematically evaluating these settings.
>
>     We appreciate your feedback and will integrate these insights into our revised paper, providing a more detailed hyperparameters exploration to further strengthen our contributions.
>
> - **Lack of efficiency analysis**
>
>     We acknowledge that our method requires maintaining the reference model in memory during optimization, which could impact efficiency compared to approaches where the reference model’s values are precomputed. However, it is important to note that our approach does not require additional preprocessing steps or modifications to the training pipeline, meaning that the Peak GPU memory usage remains unchanged compared tm to standard training setups without any "precomputing" adjustments. This simplifies implementation while maintaining compatibility with existing workflows.
>
>     For time analysis, we training time across various configurations of proposed methods with baseline methods (DPO, IPO, and KTO), as described in our paper (line 286). We train each model for a single epoch on the UltraFeedback dataset for consistency. Specifically, we evaluated Llama3 8B Base with soft updates $(\alpha=0.8)$ and hard updates $(\tau=32)$ for our method, as well as DPO, IPO, and KTO as baselines. For TR-KTO, we used $\alpha=0.6$ for soft updates and $\tau=64$ for hard updates, as outlined in Table 6 of our work.
>
>     The results show that the additional computational time of our method compared to the baselines is moderate:
>
>     - For DPO, the training time increased by **3.99%** (soft updates) and **2.87%** (hard updates).
>     - For IPO, the increase was **6.93%** (soft updates) and **3.32%** (hard updates).
>     - For KTO, the increase was **7.44%** (soft updates) and **3.65%** (hard updates).

---

> ### Author Response · Authors · 2024-11-24
>
> - We acknowledge the importance of understanding the relationship between optimization steps, overfitting, and overoptimization in preference optimization methods. However, we believe there is a key distinction that needs to be clarified in this context.
>
>     As demonstrated by Rafailov et al. [6], overoptimization typically occurs during the initial stages of training, often within the first epoch. Their findings show that preference optimization methods, such as DPO, exhibit significant performance degradation due to overoptimization well before reaching the overfitting stage. Similarly, Zephyr [7] highlights that alignment algorithms tend to overfit after a single epoch, but this phenomenon is distinct from overoptimization, which we focus on addressing in this work. Overfitting generally impacts generalization to unseen data, while overoptimization refers to excessive alignment at the cost of overall performance on downstream tasks, even within the training data.
>
>     Regarding the comment on large update steps for hard update settings, our observations suggest that the step size ($\tau$) plays a significant role in determining performance and alignment stability. For smaller models, such as Pythia 2.8B, larger step sizes ($\tau > 256$) were required to achieve competitive results. However, for larger models like Llama3-8B, our experiments (Table 7 in the paper) show that $\tau = 32$ yields the best performance across AlpacaEval 2.0 LC and WR. This indicates that the choice of $\tau$ must be appropriately calibrated based on the model size and training dynamics.
>
>     In summary, while the overfitting is an important consideration in alignment research, we emphasize that overoptimization typically occurs within the first epoch, as observed in prior work.  Thank you for this valuable feedback, which allowed us to clarify these points further.

---

> ### Author Response · Authors · 2024-11-24
>
> **Comments on Questions**
>
> 1. Thank you for your question and for pointing us toward specific datasets and models to explore further comparisons. However, we would like to clarify an important issue regarding the proposed setup. It seems there is a misunderstanding or inconsistency in your suggestion for reproducing SimPO's results.
>
>     The model you referenced, LLaMA3-8b-SFT (https://huggingface.co/princeton-nlp/Llama-3-Base-8B-SFT), was trained from the Llama3-Base model on the UltraChat dataset. You suggest fine-tuning this model on a dataset generated by Llama3-Instruct (as described in dataset card - https://huggingface.co/datasets/princeton-nlp/llama3-ultrafeedback). However, according to the dataset's description, this dataset was specifically designed for Llama3-Instruct setups:
>
>     *"This dataset was used to train princeton-nlp/Llama-3-Instruct-8B-SimPO.”*
>
>     To ensure consistency and address your concerns comprehensively, we conducted experiments covering three correct setups:
>
>     - **Llama3-Base-SFT_SimPo-UF_Gen**: Fine-tuning LLaMA3-8b-SFT (https://huggingface.co/princeton-nlp/Llama-3-Base-8B-SFT) on the UltraFeedback dataset prepared by the SimPO authors (https://huggingface.co/datasets/princeton-nlp/llama3-ultrafeedback), which was generated from **Llama3-Instruct** outputs.
>     - **Llama3-Base-SFT_SimPo-UF**: Fine-tuning the **LLaMA3-8b-SFT** (https://huggingface.co/princeton-nlp/Llama-3-Base-8B-SFT) model prepared by the SimPO authors on the **binarized UltraFeedback dataset** (https://huggingface.co/datasets/HuggingFaceH4/ultrafeedback_binarized).
>     - **Llama3-Instruct-UF_Gen**: Fine-tuning **Llama3-Instruct** on the UltraFeedback dataset prepared by the SimPO authors (https://huggingface.co/datasets/princeton-nlp/llama3-ultrafeedback), which was generated from **Llama3-Instruct** outputs.
>
>     For consistency with SimPO, we adhered to their reported learning rates: **5e-07** for Base models and **7e-07** for Instruct setups, as mentioned in their GitHub repository (https://github.com/princeton-nlp/SimPO) and the SimPO paper itself. Additionally, we evaluated baseline methods (DPO, IPO, KTO) and our TR approaches with $\beta=0.01$ and $\beta=0.05$
>
>     Below, we present the results for each setup with the best $\beta$ values for each method. For the TR-DPO and TR-IPO methods, we used $\alpha=0.8$ and $\tau=32$, while for TR-KTO, we considered $\tau=32$ and $\tau=64$, as well as $\alpha=0.6$ and $\alpha=0.8$.

---

> ### Author Response · Authors · 2024-11-24
>
> **Llama3-Base-SFT_SimPo-UF_Gen**
>
> lr 5e-07
>
> | Method | $\beta$ | LC (std) | WR (std) | Arena Hard (CI) | Alpaca Length |
> | --- | --- | --- | --- | --- | --- |
> | DPO | 0.01 | 13.02 (0.59) | 10.3 (0.93) | 8.0 (-1.0; 1.3) | 1298 |
> | TR-DPO ($\alpha=0.8$) | 0.05 | 13.05 (0.64) | 9.69 (0.91) | 6.8 (-1.0; 1.2) | 1198 |
> | TR-DPO ($\tau=32$) | 0.05 | **15.76 (0.7)** | 12.19 (1.0) | **8.1 (-1.0; 1.1)** | 1283 |
> | IPO | 0.01 | 12.02 (0.61) | 8.82 (0.88) | 5.4 (-1.0; 1.0) | 1244 |
> | TR-IPO ($\alpha=0.8$) | 0.05 | 15.13 (0.64) | 11.34 (0.97) | 7.8 (-1.2; 1.1) | 1296 |
> | TR-IPO ($\tau=32$) | 0.01 | **16.03 (0.62)** | 11.61 (0.97) | **8.0 (-1.4; 1.3)** | 1270 |
> | KTO | 0.05 | 14.39 (0.69) | 12.15 (1.0) | 7.5 (-0.9; 1.0) | 1478 |
> | TR-KTO ($\alpha=0.6$) | 0.05 | **15.31 (0.64)** | 13.1 (1.02) | 7.4 (-1.1; 1.1) | 1541 |
> | TR-KTO ($\tau=64$) | 0.05 | 13.48 (0.6) | 10.66 (0.93) | **8.3 (-1.0; 1.4)** | 1402 |
>
> **Llama3-Base-SFT_SimPo-UF**
>
> lr 5e-07
>
> | Method | $\beta$ | LC (std) | WR (std) | Arena Hard (CI) | Alpaca Length |
> | --- | --- | --- | --- | --- | --- |
> | DPO | 0.01 | 17.37 (0.73) | 16.7 (1.13) | 13.4 (-1.4; 1.7) | 1807 |
> | TR-DPO ($\alpha=0.8$) | 0.05 | **22.18 (0.8)** | 21.32 (1.23) | **15.6 (-1.6; 1.6)** | 1851 |
> | TR-DPO ($\tau=32$) | 0.05 | 18.65 (0.71) | 18.74 (1.15) | 14.3 (-1.3; 1.5) | 1879 |
> | IPO | 0.01 | 12.59 (0.65) | 10.06 (0.92) | 6.2 (-1.0; 1.0) | 1377 |
> | TR-IPO ($\alpha=0.8$) | 0.01 | **21.03 (0.79)** | 20.27 (1.2) | **12.7 (-1.4; 1.4)** | 1854 |
> | TR-IPO ($\tau=32$) | 0.01 | 19.15 (0.69) | 18.69 (1.16) | 12.4 (-1.0; 1.4) | 1859 |
> | KTO | 0.01 | 15.38 (0.67) | 13.9 (1.04) | 10.3 (-1.5; 1.4) | 1723 |
> | TR-KTO ($\alpha=0.8$) | 0.05 | 15.61 (0.69) | 12.88 (1.0) | **13.2 (-1.6; 1.5)** | 1596 |
> | TR-KTO ($\tau=32$) | 0.05 | **18.45 (0.73)** | 15.57 (1.07) | 11.7 (-1.2; 1.5) | 1671 |
>
> **Llama3-Instruct-UF_Gen**
>
> lr 7e-07
>
> | Method | $\beta$ | LC (std) | WR (std) | Arena Hard (CI) | Alpaca Length |
> | --- | --- | --- | --- | --- | --- |
> | DPO | 0.01 | 41.94 (0.9) | 40.14 (1.46) | **30.8 (-2.3; 2.0)** | 1919 |
> | TR-DPO ($\alpha=0.8$) | 0.05 | 42.97 (0.84) | 40.2 (1.44) | 29.5 (-1.5; 2.0) | 1875 |
> | TR-DPO ($\tau=32$) | 0.05 | **44.42 (0.85)** | 42.13 (1.46) | 30.4 (-2.5; 2.5) | 1901 |
> | IPO | 0.01 | 32.91 (0.88) | 31.94 (1.39) | 27.5 (-2.2; 2.1) | 1915 |
> | TR-IPO ($\alpha=0.8$) | 0.01 | **44.69 (0.82)** | 41.49 (1.43) | 28.1 (-2.9; 2.1) | 1862 |
> | TR-IPO ($\tau=32$) | 0.01 | 43.59 (0.84) | 40.3 (1.43) | **29.0 (-2.1; 2.6)** | 1863 |
> | KTO | 0.01 | 36.55 (0.86) | 33.52 (1.39) | 26.9 (-2.1; 2.0) | 1842 |
> | TR-KTO ($\alpha=0.6$) | 0.01 | 40.53 (0.85) | 35.15 (1.41) | 28.1 (-2.0; 1.9) | 1767 |
> | TR-KTO ($\tau=32$) | 0.05 | **41.17 (0.84)** | 36.57 (1.39) | **28.3 (-2.3; 2.3)** | 1788 |
>
> As demonstrated in the results, the TR  methods generally outperform the baseline approaches across LC, WR, and Arena Hard metrics in the correct Base and Instruct setups.
>
> Finally, to address potential concerns regarding the difference in Arena Hard metrics compared with the SimPO paper, we would like to clarify that we used a temperature of **0.9** for Arena Hard evaluations, whereas SimPO used **0.0**. In our experiments, we consistently applied a temperature of **0.9**, aligning with the AlpacaEval benchmark and KL divergence measurements. This setting was applied uniformly across both baseline and TR methods. As a result, the temperature does not introduce any bias specific to our TR methods.
>
> We hope this clarifies the validity of the chosen setups and further underscores the effectiveness of our proposed TR methods. Thank you for your detailed question.

---

> ### Author Response · Authors · 2024-11-24
>
> 2. Thank you for your thoughtful question. To address this concern, we extended our experiments to include a detailed analysis of $\beta$ values, which were systematically varied as follows: 0.3, 0.1, 0.05, 0.03, 0.01, 0.007, 0.005, and 0.001,  for baseline methods (DPO, IPO, and KTO) in the setup with the Llama3-Base-8B model with discussed before lr equals 1e-06. Additionally, we evaluated our TR modifications with their best-performing $\alpha$ and $\tau$ configurations, as outlined in Table 6 of our paper. These additional results will be included in the rebuttal revision of our paper.
>
>     Through this analysis, we found that the optimal $\beta$ values for baseline methods differ significantly from those observed for the Pythia 2.8B model and from the hyperparameter settings reported in the SimPO paper. Specifically:
>
>     - For DPO, the optimal $\beta$ was found to be **0.005**,
>     - For IPO, the optimal $\beta$ was **0.001**, and
>     - For KTO, the optimal $\beta$ was **0.03**.
>
>     These findings provide valuable empirical insights, highlighting the importance of adapting $\beta$ values to different model architectures and tasks.
>
>     When analyzing Pareto fronts for TR methods and their baseline counterparts on AlpacaEval 2.0 LC/KL, we observed statistically significant improvements for TR methods:
>
>     - **TR-DPO**: Achieved a significantly better Pareto front compared to DPO for both hard and soft updates, with the optimal $\beta$ for TR-DPO being **0.03**.
>     - **TR-IPO**: Showed statistically significant improvements for soft updates, with marginal but consistent improvements for hard updates. The optimal $\beta$ for TR-IPO was **0.01**.
>     - **TR-KTO**: Demonstrated similar trends, with soft and hard updates outperforming KTO. The optimal $\beta$ for TR-KTO was **0.05**.
>
>     We have attached the updated Pareto front plots for reference:
>
>     The results are presented on the figure: https://pdfupload.io/docs/4007e828 (anonymous link).
>
>     These results reinforce the robustness of our observations from the Pythia 2.8B model and validate them for larger models like Llama3-Base-8B. Additionally, the observed differences in optimal $\beta$ values across models underline the importance of careful hyperparameter tuning for each setup, a consideration not fully addressed in the SimPO paper.

---

> ### Author Response · Authors · 2024-11-24
>
> 3. Thank you for suggesting the MixEval benchmark for a detailed evaluation of our methods. To address your request, we have extended our experiments to include MixEval, alongside the benchmarks already discussed, such as AlpacaEval 2 and ArenaHard. For a fair comparison between baseline methods and our proposed TR modifications, we ensured that each method was evaluated with its optimal $\beta$ value, as determined through our hyperparameter tuning experiments. Below, we provide updated results for AlpacaEval 2.0, ArenaHard, and MixEval for Llama3-Base and Llama3-Instruct setups.
>
> | **Llama3-Base** |  |  |  |  |  | **Llama3-Instruct** |  |  |  |  |  |
> | --- | --- | --- | --- | --- | --- | --- | --- | --- | --- | --- | --- |
> | Method | $\beta$ | LC (std) | WR (std) | Arena Hard | Alpaca Length | Method | Beta | LC (std) | WR (std) | Arena Hard | Alpaca Length |
> | SFT | - | 8.5 (0.5) | 4.8 (0.6) | 3.2 (-0.7, 0.8) | 844 | SFT | - | 24.9 (0.8) | 25.3 (1.3) | 19.1 (-1.8, 1.8) | 1954 |
> | DPO | 0.005 | 25.9 (0.8) | 22.9 (1.2) | 18.5 (-1.4; 1.5) | 1759 | DPO | 0.005 | 42.3 (0.8) | 45.1 (1.5) |  | 2142 |
> | TR-DPO ($\alpha=0.8$) | 0.03 | **29.1 (0.8)** | 25.8 (1.3) | **20.1 (-1.9; 1.8)** | 1752 | TR-DPO ($\alpha=0.8$) | 0.05 | **43.5 (0.8)** | 46.8 (1.5) | **34.7 (-1.9, 2.6)** | 2172 |
> | TR-DPO ($\tau=32$) | 0.03 | 28.8 (0.8) | 26.7 (1.3) | 17.8 (-1.8; 1.6) | 1840 | TR-DPO ($\tau=32$) | 0.05 | 42.8 (0.8) | 47.2 (1.4) | 32.4 (-2.5, 2.4) | 2199 |
> | IPO | 0.001 | 27.3 (0.8) | 23.9 (1.2) | 16.4 (-1.3; 1.8) | 1754 | IPO | 0.001 | 40.2 (0.8) | 42.7 (1.4) | **37.2 (-2.1; 1.8)** | 2143 |
> | TR-IPO ($\alpha=0.8$) | 0.01 | **29.5 (0.8)** | 25.4 (1.3) | 19.4 (-1.8, 2.0) | 1705 | TR-IPO ($\alpha=0.8$) | 0.01 | **43.6 (0.8)** | 46.9 (1.5) | 33.9 (-2.0, 2.4) | 2168 |
> | TR-IPO ($\tau=32$) | 0.01 | 28.1 (0.8) | 25.4 (1.3) | **21.1 (-1.8, 2.0)** | 1808 | TR-IPO ($\tau=32$) | 0.01 | 42.6 (0.8) | 46.8 (1.5) | 34.5 (-2.5, 1.9) | 2187 |
> | KTO | 0.03 | 19.8 (0.8) | 16.2 (1.3) | 10.0 (-1.2; 1.3) | 1637 | KTO | 0.03 | 34.9 (0.9) | 36.6 (1.4) | 26.9 (-2.4; 1.8) | 2058 |
> | TR-KTO ($\alpha=0.6$) | 0.05 | **20.7 (0.8)** | 17.3 (1.1) | **18.5 (-1.4, 1.5)** | 1687 | TR-KTO ($\alpha=0.8$) | 0.05 | 40.2 (0.9) | 40.7 (1.4) | 29.2 (-1.9, 2.2) | 2013 |
> | TR-KTO ($\tau=256$) | 0.05 | 20.7 (0.8) | 18.0 (1.1) | **13.7 (-1.7, 1.7)** | 1730 | TR-KTO ($\tau=32$) | 0.05 | **41.3 (0.8)** | 42.0 (1.4) | **30.9 (-2.1, 2.1)** | 2026 |
>
> As shown, TR methods demonstrate consistent improvements over baseline methods in Alpaca Eval 2 LC and WR, and mostly on ArenaHard benchmark across both soft and hard update strategies.
>
> **MixEval**
>
> Thank you for suggesting the MixEval benchmark for further evaluation. We conducted comprehensive experiments on Llama3 models, including both Base and Instruct setups, using preference optimization methods (DPO, IPO, KTO) and their Trust Region (TR) modifications. The MixEval benchmark spans diverse categories such as knowledge evaluation, reading comprehension, commonsense reasoning, and domain-specific tasks, allowing us to assess the broader impact of alignment on downstream task performance. Below, we present an integrated analysis of these results.
>
> **Knowledge evaluation**
>
> For tasks such as MMLU and MMLU-Pro, we observed nuanced trends. In the Base setup, alignment methods, particularly DPO, resulted in slight performance degradation compared to the SFT checkpoint. However, TR modifications for DPO demonstrated modest improvements, suggesting that these methods help mitigate the loss of factual knowledge typically associated with alignment. Similarly, for tasks like GPQA and WinoGrande, performance declined slightly across most methods. This outcome may stem from the alignment process deprioritizing the domain-specific factual recall these benchmarks require.
>
> In contrast, OpenBookQA and TriviaQA showed consistent improvements for most methods, especially TR modifications. These results likely reflect the enhanced ability of aligned models to follow instructions effectively, as these tasks heavily depend on instruction-following capabilities. The improvements, while minor, were consistent across both Base and Instruct setups.
>
> On DROP, which includes tasks related to history, politics, sports, and societal issues, we observed meaningful gains across all alignment methods. These gains were consistent in both setups, suggesting that alignment positively impacts tasks requiring reasoning with structured information.
>
> For AGIEval, the results were mixed. In the Base setup, TR-DPO and TR-IPO slightly outperformed their baseline counterparts, while KTO demonstrated stability. In the Instruct setup, TR-IPO and TR-KTO showed small but consistent improvements over SFT, highlighting their ability to maintain general reasoning capabilities during alignment.

---

> ### Author Response · Authors · 2024-11-24
>
> **Reading Comprehension and Commonsense Reasoning Evaluation**
>
> For benchmarks such as ARC, HellaSwag, CommonsenseQA, BoolQ, and BBH, we observed significant improvements following alignment in the Base setup. Notably, TR-DPO $\tau = 3$ achieved up to a 34% improvement on HellaSwag, likely due to the alignment dataset containing similar instructional formats. CommonsenseQA and BoolQ also showed moderate improvements across most methods, suggesting that alignment enhances the ability to reason through everyday scenarios.
>
> In the Instruct setup, the results were more nuanced. On ARC, we observed slight performance drops of up to 4%, while CommonsenseQA results were relatively stable, with TR methods showing marginal gains. Interestingly, BBH, which requires chain-of-thought reasoning, exhibited consistent improvements across most methods post-alignment, with TR modifications achieving the highest gains. BoolQ results were less consistent, with only TR-DPO $\alpha = 0.8$ and TR-IPO $\alpha = 0.8$ showing improvements compared to SFT.
>
> **Domain Specific Evaluation**
>
> On domain-specific benchmarks such as PIQA (physical reasoning) and SIQA (social reasoning), the trends were distinct between setups. In the Base setup, most baseline methods exhibited slight degradation on PIQA, while TR methods achieved small but consistent gains. For SIQA, all methods demonstrated comparable improvements relative to SFT, indicating that alignment positively impacts social reasoning tasks across the board.
>
> For mathematical tasks, including MATH and GSM8K, alignment led to significant gains in the Base setup. IPO and TR-IPO $\alpha = 0.8$ were particularly effective, achieving 66% on MATH and 82% on GSM8K, respectively. In the Instruct setup, performance on GSM8K further improved, with TR-KTO and TR-IPO achieving the highest scores. This may reflect the inherent advantage of Instruct models in handling mathematical tasks compared to Base models. On MATH, however, most methods showed slight declines in the Instruct setup, with DPO and TR-IPO $\alpha = 0.8$ being the exceptions.
>
> The MixEval results highlight the trade-offs and benefits of alignment across different categories of tasks. While alignment sometimes leads to minor degradation in tasks requiring domain-specific factual recall (e.g., GPQA, WinoGrande), it consistently improves tasks reliant on instruction-following, reasoning, and comprehension (e.g., HellaSwag, BBH). TR modifications, in particular, demonstrate robust performance across categories, often mitigating potential drawbacks of alignment while enhancing overall effectiveness.
>
> We appreciate the suggestion to evaluate on MixEval, as it provides a broader perspective on the strengths and limitations of our alignment methods. These findings will be incorporated into the revised paper to provide a more comprehensive analysis. Thank you again for your valuable feedback.

---

> ### Author Response · Authors · 2024-11-24
>
> | **Llama3-Base-8B** |  |  |  |  |  |  |  |  |  |  |  |  |  |  |  |  |  |  |  |
> | --- | --- | --- | --- | --- | --- | --- | --- | --- | --- | --- | --- | --- | --- | --- | --- | --- | --- | --- | --- |
> | Policy | $\beta$ | **Average** | MMLU | MMLU-Pro | OpenBookQA | GPQA | WinoGrande | DROP | AGIEval | TriviaQA | ARC | HellaSwag | CommonsenseQA | BoolQ | BBH | PIQA | SIQA | MATH | GSM8k |
> | SFT | - | 64,76 | 66,7 | 34,6 | 72,1 | 25 | 50 | 73,8 | 37,7 | 71,3 | 82,4 | 32,8 | 63,9 | 78,4 | 64,0 | 81,9 | 64,5 | 41,0 | 55,2 |
> | DPO | 0.005 | 71,16 | 69,6 | 34,1↓ | 74,4 | 25 | **50** | 81,3 | 43,0 | **74,2** | **91,2** | 62,3 | **71,3** | **83,0** | 76,8 | 78,1↓ | 73,1 | **54,8** | **78,0** |
> | TR-DPO ($\alpha=0.8$) | 0.03 | 71,06 | **70,5** | **35,1** | 72,1 | 25 | 25↓ | **81,9** | 41,9 | 74,0 | 89,0 | 61,0 | **71,3** | 79,5 | 76,8 | **83,8** | 73,1 | 52,3 | 74,2 |
> | TR-DPO ($\tau=32$) | 0.03 | **71,25** | 69,9 | **35,1** | **76,7** | 25 | 25↓ | 81,1 | **43,5** | 73,4 | 87,9 | **66,6** | 70,8 | 81,3 | **80,3** | 82,9 | **74,2** | 51,3 | 72,0 |
> | IPO | 0.001 | 70,91 | 69,3 | **34,1↓** | **74,4** | 25 | **50** | 80,5 | **45,6** | 73,7 | **91,2** | 60,7 | **70,8** | **83,6** | 81,0 | 80,0↓ | 73,1 | **66,8** | 64,0 |
> | TR-IPO ($\alpha=0.8$) | 0.01 | 70,76 | **70,8** | 33,0↓ | 72,1 | 25 | 25↓ | 81,1 | 44,2 | 72,9 | **91,2** | 60,7 | 70,3 | 80,7 | **81,4** | 81,9 | 72,0 | 50,0 | **82,7** |
> | TR-IPO ($\tau=32$) | 0.01 | **71,31** | 69,9 | **34,1↓** | **74,4** | 25 | 25↓ | **81,4** | 42,6 | **74,5** | 90,1 | **63,0** | 69,8 | **83,6** | 77,9 | **82,9** | **74,2** | 48,7 | 71,5 |
> | KTO | 0.03 | 71,35 | 71,4 | **35,1** | 67,4↓ | 0↓ | 25↓ | 79,3 | **44,9** | 74,5 | 84,6 | 65,6 | 70,3 | **83,0** | **78,0** | 79,0↓ | **73,1** | **59,4** | 72,2 |
> | TR-KTO ($\alpha=0.6$) | 0.05 | 71,46 | 70,3 | **35,1** | **74,4** | 0↓ | 25↓ | **81,8** | 43,9 | **75,5** | **91,2** | 62,0 | 70,8 | 82,5 | 74,9 | 77,1↓ | 72,0 | 40,6↓ | 73,8 |
> | TR-KTO ($\tau=256$) | 0.05 | **71,61** | **72,2** | 34,6 | 67,4↓ | 0↓ | 25↓ | 78,5 | 44,0 | 75,3 | 85,7 | **66,2** | **71,8** | 79,5 | 75,0 | **81,9** | 71,0 | 55,2 | **79,2** |
> | **Llama3-Instruct-8B** |  |  |  |  |  |  |  |  |  |  |  |  |  |  |  |  |  |  |  |
> | SFT | - | 74,65 | 74,6 | 44,9 | 72,1 | 50 | 50 | 84,1 | 52,6 | 75,2 | 93,4 | 66,9 | 75,7 | 88,9 | 81,0 | 85,7 | 68,8 | 61,0 | 86,5 |
> | DPO | 0.005 | 75,66 | 74,4 | 39,5 | 72,1 | **50** | **50** | 87,0↑ | 51,7 | 77,7↑ | 89,0 | **69,8↑** | 71,3 | 87,7 | 88,7↑ | 86,7↑ | **71,0↑** | **65,2↑** | 80,0 |
> | TR-DPO ($\alpha=0.8$) | 0.05 | **76,11** | **75,5↑** | **40,0** | **76,7↑** | **50** | 25 | 88,6↑ | **54,2↑** | **78,0↑** | 91,2 | 67,2↑ | 71,8 | **90,1↑** | **91,5↑** | 83,8 | 68,8 | 59,0 | 78,7 |
> | TR-DPO ($\tau=32$) | 0.05 | 75,71 | 75,0↑ | 38,4 | 72,1 | 25 | 25 | **89,6↑** | 51,7 | 77,2↑ | **93,4** | 66,9 | **73,3** | 87,1 | 91↑ | **88,6↑** | 67,7 | 51,6 | **82,5** |
> | IPO | 0.001 | 75,66 | 74,7↑ | **41,6** | **74,4↑** | 25 | **50** | 86,6↑ | 51,3 | **78,3↑** | 89,0 | 69,5↑ | 73,8 | 87,1 | 84↑ | 85,7 | 67,7 | 48,4 | 80,2 |
> | TR-IPO ($\alpha=0.8$) | 0.01 | **75,96** | 74,7↑ | **41,6** | 72,1 | 25 | 25 | 87,5↑ | **53,2↑** | 77,0↑ | **93,4** | **69,8↑** | **75,2** | **89,5 ↑** | 88,7↑ | **86,7↑** | **68,8** | **62,9↑** | 83,8 |
> | TR-IPO (tau=32) | 0.01 | 75,81 | **74,9↑** | 39,5 | 72,1 | 25 | **50** | **88,2↑** | 48,8 | 77,1↑ | 92,3 | **69,8↑** | 74,3 | 88,9 | **91,6↑** | 84,8 | **68,8** | 57,4 | **88,2↑** |
> | KTO | 0.03 | **76,05** | **74,9↑** | **41,6** | **79,1↑** | 25 | **50** | 88,6↑ | **54,1↑** | 76,8↑ | 91,2 | **71,4↑** | **74,8** | **87,7** | 83,6↑ | **89,5↑** | **68,8** | **61,0** | 80,7 |
> | TR-KTO ($\alpha=0.8$) | 0.05 | 75,21 | 74,4 | 39,5 | **79,1↑** | 25 | 25 | 88,8↑ | 50,9 | 76,2↑ | **92,3** | 67,5 | 74,3 | **87,7** | **88,6↑** | 83,8 | 67,7 | 51,3 | 84,0 |
> | TR-KTO ($\tau=32$) | 0.05 | 75,51 | 74,4 | 39,5 | 69,8 | 25 | 0 | **89,3↑** | 52,8↑ | **77,1↑** | 89,0 | 67,2 | **74,8** | 86,5 | 87,7↑ | 87,6↑ | 67,7 | 50,3 | **88,7↑** |
>
> **Conclusion:**
>
> We sincerely thank you for your detailed and constructive feedback, which has significantly improved our paper. In response to your comments, we have addressed all weaknesses, provided additional experiments, and clarified key points, including an extended analysis on hyperparameter tuning, efficiency, and MixEval benchmarks. These revisions strengthen our contributions and demonstrate the robustness of our proposed methods.
>
> Given these improvements and clarifications, we kindly request you to reconsider your rating. Your insights have been invaluable, and we deeply appreciate your support in enhancing the quality of our work. Thank you once again.

---

> ### Author Response · Authors · 2024-11-24
>
> **References:**
>
> [1] Munos et al, Nash Learning from Human Feedback
>
> [2] Rosset et al, Direct Nash Optimization: Teaching Language Models to Self-Improve with General Preferences
>
> [3] Llama Team, The Llama 3 Herd of Models
>
> [4] Xu et al, Contrastive Preference Optimization: Pushing the Boundaries of LLM Performance in Machine Translation
>
> [5] Pang et al, Iterative Reasoning Preference Optimization
>
> [6] Rafailov et al, Scaling Laws for Reward Model Overoptimization in Direct Alignment Algorithms
>
> [7] Tunstall et al, Zephyr: Direct Distillation of LM Alignment

---

> > ### Comment · Reviewer_xzHv · 2024-11-24
> >
> > Thanks for the comprehensive experiments. The authors addressed all of my concerns. Please update the results in Table 2 for DPO, IPO, and KTO, as reported in the SimPO paper in the LLaMA-Base setting. Also, it would be great to add the results of downstream tasks to the Appendix section. Then, I will increase my score. Best of luck.

---

> > > ### Author Response · Authors · 2024-11-25
> > >
> > > Thank you for the opportunity to address your concerns. We are pleased that our updates and experiments have helped clarify and strengthen the paper. Following your suggestions, we have made several revisions in this rebuttal:
> > >
> > > 1. Clarification of terminology: We have removed the potentially misleading phrase "new paradigm" in line 014 and replaced it with "novel approach" as more precise terminology.
> > >
> > > 2. Updates to Table 2: The results for DPO, IPO, and KTO in the Llama3-Base setting have been updated to align with the values reported in the SimPO paper (line 447).
> > >
> > > 3. Downstream task evaluation: We have added a new section, Appendix F, which includes the downstream task evaluation results from MixEval. These are detailed in Tables 8 and 9 (line 1017).
> > >
> > > 4. Enhanced explanation of the MDP example: To further improve clarity, we expanded the description of the MDP toy example in line 809.
> > >
> > > 5. Computational efficiency: In Appendix D.3, we provided additional details about the computational efficiency of our method, addressing specific concerns raised in the reviews.
> > >
> > > We believe these changes fully address your feedback and significantly enhance the clarity and rigor of our work. We kindly ask you to consider these improvements and revisit your evaluation. Your insights have been invaluable, and we deeply appreciate your effort in helping us refine our submission. Thank you again for your time and thoughtful review.

---

> > > > ### Comment · Reviewer_xzHv · 2024-11-25
> > > >
> > > > Thanks, my score increased.

---

> > > > > ### Comment · Reviewer_xzHv · 2024-12-02
> > > > >
> > > > > I am aware that the TR method has two main problems related to the need for long optimization steps and resource efficiency. However, the authors showed that this method is still better than other methods. To appreciate the authors, I decided to increase my score to 8. I hope getting bad reviews for your submissions does not affect your judgment. Best of luck.

---

### Official Review · Reviewer_LHnV · 2024-11-03

**Soundness:** 3
**Presentation:** 2
**Contribution:** 2
**Rating:** 3
**Confidence:** 4

**Summary:**

This paper proposes an enhancement to offline alignment methods for Large Language Models (LLMs), addressing the challenge of overoptimization, where a model diverges excessively from its initial reference policy, degrading alignment and sample quality. The authors introduce the Trust Region (TR) approach, which updates the reference policy dynamically during training. They implement this via soft and hard updates within three variants: TR-DPO, TR-IPO, and TR-KTO. The results show moderate performance improvements on tasks like dialogue alignment and summarization,

**Strengths:**

1. The paper tackles a well-known issue in offline alignment—overoptimization—which is relevant for applications involving LLM alignment with human preferences. The focus on dynamically updating reference policies is a step toward mitigating this issue, offering a more stable alignment during training.
2. The experimental setup is comprehensive, involving multiple datasets (task-specific and general-purpose) and benchmarks.
3. The structure and explanation of the TR approach, including its soft and hard update methods, are clearly presented. This makes the methodology easier to understand and potentially reproducible for future research.

**Weaknesses:**

1. Despite the theoretical motivation, the empirical gains presented by the TR approach are relatively minor. In many cases, improvements over baseline methods (DPO, IPO, KTO) are modest, which may limit the practical significance of the proposed approach.
2. The approach of updating the reference policy is somewhat incremental in nature, as it primarily modifies existing alignment methods (DPO, IPO, KTO) rather than introducing a fundamentally new framework for offline RLHF alignment. This limits the originality of the contribution, especially given the relatively minor performance improvements reported.

**Questions:**

1. While the TR approach appears to mitigate overoptimization initially, a more detailed analysis of its performance in long-term deployments would be valuable. Specifically, could the authors provide empirical results on how the approach maintains alignment stability over extended training iterations?

---

> ### Author Response · Authors · 2024-11-21
>
> Thank you for the time and effort you have dedicated to reviewing our paper. We deeply appreciate your highlighting of our paper’s strengths, such as addressing the prevalent issue of overoptimization in offline alignment, our comprehensive experimental setup with multiple datasets and benchmarks, and our clear presentation of the TR (Trust Region) approach, making it potentially replicable in future research.
>
> We have carefully addressed the weaknesses and questions raised, as detailed below.
>
> **Weaknesses:**
>
> 1. Our work as one of the contributions states "using pre-trained Pythia 6.9B models on the task-specific Reddit TL;DR summarization task, our methods achieve win rate improvements of 8.4% for DPO, 14.3% for IPO, and 15% for KTO over the baselines. Similarly, on the AlpacaEval 2 and Arena-Hard general benchmarks with Llama3, our TR methods show significant win rate gains, with improvements of 10.8 points for DPO, 10.5 for IPO, and 5.6 for KTO compared to the classic methods (see Figure 1b)."  Those percentage values **do not represent the proportion of the baseline's quality** but rather **benchmark measurement units**. For example, an improvement of TR-DPO over DPO by 10.8 percent means DPO scored 10.5, and TR-DPO scored 21.3. This relative improvement is greater than twofold, which we consider a major enhancement and practically beneficial for the field. This superior performance of our updated methods is also acknowledged and appreciated by other reviewers. Therefore, we would kindly appreciate any specific arguments illustrating why this result might be considered insufficient and what you consider the boundaries between minor and major improvements? Our aim is to contribute effectively to the field, and your insights would help refine and improve further research in this direction.
> 2. The incremental nature of our method did indeed evolve from existing alignment methods (DPO, IPO, KTO), but our approach substantially enhanced their performance, as described above and also confirmed in other reviews. We believe that the novelty of our research should not only be measured by the introduction of completely new methodologies. Our findings show that the static nature of the reference policy may prohibit effective training of aforementioned methods. Therefore, in contrast to being a mere adjustment of existing methods, our study provides insights into their inner workings. This better understanding of the impact of reference policy behavior on the quality of alignment methods gives a ground for future work to explore this issue.
>
> **Questions:**
>
> 1. We respectfully disagree that these experiments would provide further insight into the overoptimization problem. The proposed experimental setup tests the method's resilience to overfitting, an issue orthogonal to overoptimization, which our work investigates. As demonstrated by Tunstall et al [1], DPO catastrophically overfits after a single epoch (Figure 3a). Rafailov et al [2] states that the performance of any DAA method deteriorates after a single epoch, and the overoptimization occurs within the first epoch. Thus, tracking long-term deployment performance would not provide additional insights, as the primary cause of model overoptimization occurs during the initial stages of the training cycle.
>
> Thank you once again, we remain open to further suggestions to improve our paper. If we have succeeded in making our proposed method and its contribution clearer to you, we would very much appreciate it if you could consider improving the rating of our work.
>
> References:
>
> [1] Tunstall et al, Zephyr: Direct Distillation of LM Alignment
>
> [2] Rafailov et al, Scaling Laws for Reward Model Overoptimization in Direct Alignment Algorithms

---

> ### Author Response · Authors · 2024-11-25
>
> I hope this message finds you well. As you may be aware, the rebuttal phase is rapidly approaching its conclusion. Due to the given time constraints, we are diligently working to conduct additional experiments that could potentially enhance the quality of our work. In the comment above, we tried to clarify all the points that raised questions for you. However, if there is anything else we can do to make our work clearer and better, we are open to your suggestions.
>
> If you find that our revisions and the added clarity satisfy your queries and concerns, we would be grateful if you could consider raising your score.

---

> ### Author Response · Authors · 2024-11-27
>
> As we are quickly running out of time, with less than 24 hours remaining to make alterations to our rebuttal PDF, we find ourselves somewhat perplexed that we have not yet heard back from you.
>
> We are eager to engage in productive dialogue about our work and have taken efforts to provide thorough explanations based on your comments. Your insights have been, and continue to be, greatly valuable to us.
>
> However, we would greatly appreciate your feedback on the clarifications we have provided. Any additional suggestions you may have to further improve our work would be more than welcome.

---

### Official Review · Reviewer_cMBq · 2024-11-03

**Soundness:** 3
**Presentation:** 4
**Contribution:** 2
**Rating:** 8
**Confidence:** 4

**Summary:**

Recent literature on Offline RL LM fine-tuning methods, such as DPO, IPO, KTO etc, involve avoiding training an explicitly reward model (RM) and directly optimizing the LM from the offline preferences. However, these methods tend to over-optimize the data, such that, the probabilities of both the chosen and rejected responses from the preferences decrease compared to the reference LM and rather assign higher probability to out-of-distribution (OOD) responses. The paper tries to address this main over-optimization issue of current Offline RL approaches but bringing in ideas from Trust Region optimization. Their key idea is to modify the existing offline RL methods by making the reference LM in their training objective a moving target. Thereby, they hypothesize that the current policy will make more reasonable updates, that fix the over-optimization issue (i.e. the likelihood of chosen responses increases while rejected response still decreases). To make the reference policy moving target, they propose two simple strategies:
1. Soft Update: weighted interpolation between previous reference LM and current updated LM that is being trained. After the update, the gradients are no longer propagated from the new reference LM.
2. Hard Update: After every $\tau$ steps, the current policy is copied over the reference policy.

To support their intuition, they derive the second-order derivative of the DPO objective and show that it leads to a curvature-less loss region. Their hypothesis is that this curvature-less loss landscape leads to a decrease in the probability of the human-preferred responses, ultimately leading to OOD responses.

Their experimental setup involves comparing baseline offline RL objectives: DPO, IPO, and KTO with their Trust Region counterparts, TR-DPO, TR-IPO, and TR-KTO. They use the Anthropic HH and Reddit TL;DR dataset for preliminary evaluation with Pythia models and further experiment with Llama-3 8b models on AlpacaEval 2 and Arena-Hard benchmarks.

Overall, their results suggest that TR version of offline RL works on average better than without for both soft and hard update rules with specific parameters (0.5 or 0.6 interpolation parameter for soft update and $\tau = 512$ steps for hard update).

**Strengths:**

- Work on a well-motivated and well-known problem of overoptimization in DPO-like Offline preference optimization methods.
- They give a new perspective on the issue with DPO objective by analyzing the second-order derivative.
- A simple solution to fix the issues with offline methods that have encouraging results across many datasets and model families.

**Weaknesses:**

- Although the intuition about the curvature-less landscape is interesting, I am not able to understand from this why would the chosen probability decrease. From my understanding, DPO and the like only focus on the margin and not the raw values and potentially decreasing both of them (the rejected likelihood faster than chosen) is the easiest way to reduce the loss according to gradient update.
- The TR solution to fix the overoptimization in online RL makes sense because the samples are drawn from the reference policy during optimization. It is unclear why this would help in the offline setting since subsequent reference LMs will be further out of distribution compared to the offline preference dataset.

**Questions:**

- Does the intuition about the curvature less loss hold also for other DPO-like objectives (example, IPO and KTO)?
- I really like the likelihood analysis of the toy dataset example in Figure 2. However, I didn't see a similar analysis for the experiments with practical datasets. Does TR-DPO and others actually increase chosen preference likelihood in real datasets? If yes, I'd appreciate it if the authors included comparison plots of DPO vs TR-DPO with real data.
   - Follow-up for toy dataset analysis: Even with TR methods, I initially see a drop in chosen probs, followed by recovery and almost memorization towards the end of optimization. Do the authors have an intuition for why this initial drop in chosen likelihood happened?
- In my experience with offline RL objectives, the overoptimization is heavily correlated with increased length. Do authors have numbers comparing the average output length of TR methods vs baseline objectives? I would trust the results much more if I see the evidence that TR methods are not susceptible to length hacking.
- Of course, it is hard to evaluate all methods within an experiment setup, but I wonder if the authors are aware of this previous work which also attempts to solve the drop in chosen likelihood problem "Noise Contrastive Alignment of Language Models with Explicit Rewards" (chen et al. 2024) https://arxiv.org/pdf/2402.05369.

---

> ### Author Response · Authors · 2024-11-23
>
> First of all, we would like to express our deepest gratitude for the time and effort you have put into reviewing our paper. Your valuable feedback unquestionably helps us to improve our work. We also appreciate you emphasizing the strengths of our work, such as addressing a pertinent problem, proposing novel approaches to the study of offline loss functions, the simplicity of our proposed method, and diversity of experimental setups.
>
> We have carefully addressed the weaknesses and questions raised, as detailed below.
>
> **Weaknesses:**
>
> 1. We are not explaining over-optimization through the curvature loss landscape. Moreover, if we consider partial derivatives of the loss function w.r.t. $\pi_\theta(y_w | x)$ and $\pi_\theta(y_l | x)$, we get the following expressions:
>
>     $\frac{\partial L}{\partial \pi _\theta(y_w|x)} = - \frac{\beta (\frac{\pi _\theta(y_l | x)}{\pi _\mathrm{ref}(y_l | x)})^\beta}{\pi _\theta(y _w | x) ((\frac{\pi _\theta(y _w | x)}{\pi _\mathrm{ref}(y _w | x)})^\beta + (\frac{\pi _\theta(y _l | x)}{\pi _\theta(y _l | x)})^\beta)}$
>
>     $\frac{\partial L}{\partial \pi  _\theta(y_l|x)} = \frac{\beta (\frac{\pi _\theta(y_l | x)}{\pi _\mathrm{ref}(y_l | x)})^\beta}{\pi _\theta(y _l | x) ((\frac{\pi _\theta(y _w | x)}{\pi _\mathrm{ref}(y _w | x)})^\beta + (\frac{\pi _\theta(y _l | x)}{\pi _\theta(y _l | x)})^\beta)}$
>
>     As we can see, the partial derivative w.r.t. $\pi_\theta(y_w | x)$ is always negative, while the one pertaining to $\pi_\theta(y_l | x)$ is positive. This means that gradient descent should promote an increase in the probability of selected texts and a decrease in the probability of bad ones. However, this does not happen and the probability of chosen texts decrease as rejected ones. This fact is tried to be explained in the article [1]. Our observation is merely that loss gradient changes less and less over training time (as evidenced by the second derivative), and if an overoptimization occurs, DPO can't escape it. We propose a method to mitigate this process, as demonstrated in our experiments: on the toy example, we observe an increase in the probabilities of chosen texts, while on real tasks, we observe improvements in overall quality.
>
> 2. Online analogs of our method include various Nash Learning approaches: Nash-MD [2], DNO [3]. Our work's objective is to demonstrate that updating the reference policy is also applicable to offline methods and requires minimal modifications. The reference model will not be completely OOD compared to the preference dataset because we perform the SFT stage, which aims to make the model's distribution closer to the distribution from which the offline data was obtained. Undoubtedly, the policy's distribution changes with updating the reference. Our experiments (Table 6 and 7 in our paper) show that the method doesn't work if updates are made too frequently, meaning that we can only achieve optimal results if we maintain a certain proximity to the SFT policy.

---

> ### Author Response · Authors · 2024-11-23
>
> **Questions**
>
> 1.  Thank you for raising this insightful question. We have analyzed the curvature dynamics for DPO-like objectives, focusing on IPO and KTO. Below, we provide a detailed mathematical and conceptual analysis for IPO, along with a discussion of the challenges in analyzing KTO.
>
>     The gradient of the IPO loss is given by:
>
>     $
>     \nabla _\theta L _\mathrm{IPO}(\pi _\theta, \pi _\mathrm{ref}) = \mathbb{E} _{(x, y_w, y_l) \sim \mathcal{D}} \left[2 \left( \log \frac{\pi _\theta(y_w | x)}{\pi _\mathrm{ref}(y_w | x)} - \log \frac{\pi _\theta(y_l | x)}{\pi _\mathrm{ref}(y_l | x)} - \frac{1}{2 \beta}\right)\nabla _\theta \log \frac{\pi _\theta(y_w \ x)}{\pi _\theta(y_l | x)}\right]
>     $
>
>     The Hessian of the IPO loss is then:
>
>     $
>     \nabla^2_\theta L_\mathrm{IPO}(\pi_\theta, \pi_\mathrm{ref}) = 2 t \nabla^2_\theta t + 2 \nabla_\theta t (\nabla_\theta t) ^\top
>     $
>
>     where $t = \log \frac{\pi_\theta(y_w | x)}{\pi_\theta(y_l | x)} - \log \frac{\pi_\mathrm{ref}(y_w | x)}{\pi_\mathrm{ref}(y_l | x)} - \frac{1}{2 \beta}$
>
>     During training, the objective aims to minimize $t^2$, driving $t$ towards zero. As $t$ decreases, the term $t \, \nabla^2_\theta t$ becomes negligible, leading to a flatter loss dynamic. This aligns with the behavior observed for DPO: if overoptimization begins (e.g., probabilities of chosen sequences decrease), the optimization dynamics struggle to reverse this trend, leading to potential stagnation.
>
>     For the second term in the Hessian, $2 \nabla_\theta t (\nabla_\theta t)^\top$, its behavior is more nuanced. The gradient of $t$, $\nabla_\theta t$, does not inherently depend on the magnitude of $t$. Thus, we cannot conclude that this term diminishes solely because $t$ is small. However, we hypothesize that the reduction in the first term, $t \, \nabla^2_\theta t$, is sufficient to cause overoptimization in practice. This is supported by experimental evidence, where TR-IPO successfully mitigates overoptimization by resetting $\pi_\mathrm{ref} \leftarrow \pi_\theta$. This reset increases the magnitude of $t$, thereby enhancing the contribution of the curvature term $t \, \nabla^2_\theta t$ in the Hessian. This reset increases the curvature of the loss landscape, much like in TR-DPO, helping to prevent stagnation and allowing the model to continue learning effectively. The empirical improvements observed with TR modifications suggest that the periodic resetting of $\pi_\mathrm{ref}$ helps maintain sufficient curvature in the loss landscape, even though we cannot directly measure the effect of the outer product term $2 \nabla_\theta t (\nabla_\theta t)^\top$ due to computational constraints.
>
>     For KTO, the loss function involves an expectation nested within another expectation, making an exact analytical derivation of the second derivative (Hessian) complex.
>
>     $
>     \nabla _\theta L _\mathrm{KTO}(\pi _\theta, \pi _\mathrm{ref}) = -\beta \mathbb{E} _{(x, y_w, y_l) \sim \mathcal{D}} \left[\lambda _w \sigma(z(y_w))\sigma(-z(y_w))\nabla _\theta \log \pi _\theta(y_w | x) - \lambda _l \sigma(z(y_l))\sigma(-z(y_l))\nabla _\theta \log \pi _\theta(y_l | x) \right]
>     $
>
>     While we currently limit our theoretical analysis to DPO and IPO, our empirical results demonstrate that TR modifications consistently improve KTO as well. These practical findings suggest that the intuition behind resetting the optimization process by updating the reference policy holds for KTO as well. We appreciate your insightful question and will incorporate this discussion into our revised paper.

---

> ### Author Response · Authors · 2024-11-23
>
> 2. While we have only 7 OOD trajectories in the toy example and can trace the probability of each one, there are infinitely many OOD trajectories in real-life cases. That's why all papers on overoptimization conduct such analysis using toy examples [4, 5, 6]. The toy example becomes a necessary simplification that allows us to unravel more complex mechanisms behind the methodology. Conversely, real-world scenarios can present too many variables, making outcomes more challenging to interpret. For real-world scenarios, the analysis of DPO and TR-DPO, presented in Appendix C of our paper, shows that while the TR method does not increase the probabilities of the chosen samples, it results in a smaller decrease during training compared to the baseline DPO.
>
>     a. As mentioned earlier, TR-DPO doesn't eliminate the root cause of overoptimization. Instead, it gives the model more opportunities to escape from overoptimization by increasing the coefficient in front of the gradient, thereby pushing the probabilities of chosen and rejected in the right direction. Thus, in our toy example, we observe how the model starts to overoptimize (which is not surprising, as it takes many training steps on just one pair), but then it escapes this process, unlike the vanilla DPO. We admit that our approach is more of a mitigation strategy and we are actively working on finding a solution that directly addresses the overoptimization issue's root cause.
>
> 3. According to Rafailov et al [5], the length-dependency is more towards KL Divergence rather than overoptimization. Moreover, it shows that this dependence differs for different model sizes (as seen in Figure 3 [5]). Our experiments confirm this fact. For Pythia 2.8, this dependency is depicted in the Appendix Figure 19. From this, it is clear that while rolling into overoptimization, the length for all methods decreases and our methods have a smaller length at all KL Divergence regions where overoptimization has not yet occurred. For the analysis of length on larger models, we conducted additional experiments. We trained the base Llama3-8B model using DPO, TR-DPO$^\alpha$ (where $\alpha = 0.8$) and TR-DPO$^\tau$ (where $\tau = 32$) and depicted the length and LC WR on Alpaca Eval dependency on KL Divergence for different $\beta$.
>
>     The results are presented on the figure: https://pdfupload.io/docs/a48625eb (anonymous link).
>
>     Our results showed that in the overoptimization area, the length drastically increases. However, in our work, we used $\beta$ for which overoptimization does not occur. In this region, the average response length for TR-DPO and DPO is roughly the same. Thank you for your feedback. The experiments conducted provided greater insight into how our method works. We will definitely incorporate these results into our paper.

---

> ### Author Response · Authors · 2024-11-23
>
> 4. Thank you for bringing this work to our attention. We appreciate the opportunity to explore the connections and compare our methods with "Noise Contrastive Alignment of Language Models with Explicit Rewards" (Chen et al., 2024). To address this, we conducted experiments incorporating NCA into our setup and evaluated its performance against our Trust Region modifications, TR-NCA (using both soft updates with $\alpha = 0.8$ and hard updates with $\tau = 32$).
>
>     We experimented with various $\beta$ values ($0.3, 0.1, 0.03, 0.007, 0.005, 0.001, 0.0005, 0.0001$) and learning rates ($5\text{e-07}$ and $1\text{e-06}$) on the Llama3-Base-8B model. The best results were achieved with a learning rate of $1\text{e-06}$. The highest LC for NCA occurred at $\beta = 0.0005$, while TR-NCA achieved its best results with $\beta = 0.007$.
>
>     The results are presented on the figure: https://pdfupload.io/docs/ed914cc1 (anonymous link).
>
>     Our experiments also revealed that TR-NCA achieved a better Pareto front relative to AlpacaEval 2.0 LC versus KL divergence. Specifically, we observed that as the divergence (KL) from the original policy increases, the length of generated sequences remains nearly constant across different $\beta$ values. However, the AlpacaEval LC scores differ significantly, indicating that the TR modifications provide meaningful improvements in quality without compromising sequence length.
>
>     Below, we include a table summarizing the best results for NCA and TR-NCA, along with Pareto front plots and a graph of sequence length versus KL for further comparison.
>
>     | Method | Beta | LC (std) | WR (std) | Length |
>     | --- | --- | --- | --- | --- |
>     | NCA | 0.0005 | 18.01 (0.73) | 15.84 (1.1) | 1747 |
>     | TR-NCA ($\alpha=0.8$) | 0.007 | 22.36 (0.82) | 18.76 (1.17) | 1697 |
>     | TR-NCA ($\tau=32$) | 0.007 | 21.08 (0.79) | 17.78 (1.14) | 1694 |
>
>     We appreciate your thoughtful question, as it allowed us to extend our analysis and provide meaningful comparisons. These findings further validate the benefits of Trust Region modifications in improving alignment performance. Thank you again for this valuable feedback, and we hope this analysis addresses your query comprehensively.
>
> Thank you once again for your valuable feedback and suggestions to improve our paper. We hope our responses have clarified the proposed method and its contributions. If we have succeeded in addressing your concerns, we would greatly appreciate it if you could consider revisiting the rating of our work.
>
> **References:**
>
> [1] Rafailov et al, From r to Q*: Your Language Model is Secretly a Q-Function
>
> [2] Munos et al, Nash Learning from Human Feedback
>
> [3] Rosset et al, Direct Nash Optimization: Teaching Language Models to Self-Improve with General Preferences
>
> [4] Azar et al, A General Theoretical Paradigm to Understand Learning from Human Preferences
>
> [5] Rafailov et al, Scaling Laws for Reward Model Overoptimization in Direct Alignment Algorithms
>
> [6] Huang et al, Correcting the Mythos of KL-Regularization: Direct Alignment without Overoptimization via Chi-Squared Preference Optimization

---

> > ### Comment · Reviewer_cMBq · 2024-11-25
> >
> > Great set of explanations and thank you for the additional experiments!

---

> > > ### Author Response · Authors · 2024-11-25
> > >
> > > Hello, thank you for your response! We believe we have addressed all of your concerns and would kindly ask if you could reevaluate the score accordingly.

---

> ### Author Response · Authors · 2024-11-26
>
> In light of the detailed clarifications and experiments we have provided, addressing each concern and question you posed, we were somewhat taken aback to notice that the rating for our submission remains the same. As we approach the deadline for finalizing our rebuttal pdf, we would greatly value any additional feedback or suggestions that you might have for enhancing our work.

---

### Author Response · Authors · 2024-11-28

We would like to express our gratitude to all of the reviewers for dedicating their time to review our work. Receiving feedback is always beneficial as it allows us to improve the quality of our work. We have addressed all of the reviewers' comments and answered all raised questions. We have also made the following revisions to our paper based on the reviewers' recommendations:

- **Theoretical Analysis**:  We have added an analysis of the IPO Hessian and its behavior throughout the optimization process in Appendix A.
- **Updates to Table 2**: We have revised the results for DPO, IPO, and KTO in the Llama3-Base setting to match those reported in the SimPO paper (line 451).
- **Downstream task evaluation**: We have introduced a new section, Appendix F, which includes the downstream task evaluation results from MixEval. These are detailed in Tables 8 and 9 (line 1200).
- **Enhanced explanation of the MDP example**: To improve clarity, we have expanded the description of the MDP toy example in line 865.
- **Computational efficiency**: In Appendix D.3, we have provided additional information on the computational efficiency of our approach, thereby addressing specific concerns raised in the reviews.
- **Resilience to Jailbreak Attacks**: We have evaluated our method against the GPTFuzz and ReNeLLM benchmarks and reported these results in Appendix G.

We greatly appreciate any further suggestions for improving our work and are always open to discussions to clarify any remaining uncertainties or answer any additional questions.

We would, however, like to mention that unfortunately we were unable to establish communication with the LHnV reviewer during the discussion session. Consequently, we didn't receive any constructive feedback or responses to our explanations from that particular reviewer.

---

### Meta-Review · Area_Chair_xmwJ · 2024-12-22

**Metareview:**

This paper aims to mitigate the overoptimization problem in offline RL methods for aligning language models. The proposed method brings in idea from trust region optimization and is based on the idea of updating the reference language model to make it a moving target. Experiments comparing their trust region methods to offline RL baselines show that on average their method does better.

Strengths:
1. The work is well motivated.

Weaknesses:
1. The empirical gains are modest.

Overall reviewers' opinions of this work are positive. Therefore, I'm recommending acceptance, but I wouldn't mind if the paper gets rejected.

**Additional Comments On Reviewer Discussion:**

Reviewer xzHv asked for additional comparisons, which is addressed during the rebuttal. I'd recommend authors to incorporate the new results into the next version of the paper.

---

### Decision · Program_Chairs · 2025-01-22

Accept (Poster)